# Misconceptions of Antibiotics as a Potential Explanation for Their Misuse. A Survey of the General Public in a Rural and Urban Community in Sri Lanka

**DOI:** 10.3390/antibiotics11040454

**Published:** 2022-03-27

**Authors:** Yasodhara Deepachandi Gunasekera, Tierney Kinnison, Sanda Arunika Kottawatta, Ayona Silva-Fletcher, Ruwani Sagarika Kalupahana

**Affiliations:** 1Department of Veterinary Public Health and Pharmacology, Faculty of Veterinary Medicine and Animal Science, University of Peradeniya, Peradeniya 20400, Sri Lanka; sandavphp@gmail.com (S.A.K.); ruwanikalupahana@yahoo.com (R.S.K.); 2Royal Veterinary College, University of London, London NW1 0TU, UK; tkinnison@rvc.ac.uk (T.K.); asilvafletcher@rvc.ac.uk (A.S.-F.)

**Keywords:** antibiotics, antimicrobial resistance, urban community, rural community, knowledge, awareness, perceptions, Sri Lanka

## Abstract

Reducing the growth of antimicrobial resistance (AMR) through public understanding is a goal of the World Health Organization. It is especially important in countries where antibiotics are widely available for common ailments without prescription. This study assessed understanding of antibiotics and AMR alongside perception of antibiotic usage among the general public in two diverse Sri Lankan communities: ordinary urban and indigenous rural. A cross-sectional questionnaire survey was conducted, gaining 182 urban and 147 rural responses. The majority of urban respondents (69.2%) believed that they had very good or good knowledge about antibiotics compared to 40.1% of rural respondents. Belief about knowledge and actual knowledge (measured via a test question) were correlated (r = 0.49, *p* = 0.001) for rural respondents, but not for urban respondents. Several misconceptions about antibiotics were highlighted, including that Paracetamol, a painkiller, was thought to be an antibiotic by more than 50% of both urban and rural respondents. In addition, 18.5% of urban and 35.4% of rural participants would keep and re-use what they perceived as leftover antibiotics. It is urgent that we pay attention to educating the general public regarding the identified misconceptions of these powerful drugs and their appropriate use.

## 1. Introduction

Antimicrobial resistance (AMR) has been identified as a global health problem. At present, AMR is rising at an alarming level, significantly increasing the morbidity and mortality rate worldwide [1]. Hence, it was identified by the World Health Organization (WHO) as a major public health challenge that could become the next global pandemic [2]. Furthermore, the WHO has announced that AMR could lead to 10 million deaths worldwide by 2050 and has predicted that most deaths will be in the Asian Pacific subcontinent, where Sri Lanka is located [3]. AMR has also been predicted to cause 24 million people to go into extreme poverty in the year 2030 [2].

Inappropriate and irrational use of antibiotics plays a key factor in microbes gaining resistance and triggering the current situation of AMR [4]. These concerns prompted the WHO to announce a global action plan on AMR in 2015 [5]. The first objective was to improve awareness and understanding of AMR all over the world via effective communication, education, and training. To fulfill this objective, WHO conducted a multi-country public awareness survey about AMR [6]. In addition, several researchers have investigated the knowledge, awareness, and perception (KAP) about antibiotics and AMR among communities in different parts of the world, as summarized within a systematic review by Kosiyaporn et al. [7]. The studies revealed a widespread occurrence of self-medication and purchase of antibiotics over the counter [8], the fact that many believe that humans may themselves become resistant to antibiotics [9], that many people stop taking prescribed antibiotics when they feel better [10], and that often the public re-use leftover antibiotics [11,12].

Following the WHO action plan on AMR, Sri Lanka launched a strategic plan to tackle AMR [8]. Improving awareness and understanding of AMR through effective communication in the Sri Lankan population was the first published strategy [8]. Further, two specific objectives were identified under this strategy; the first was to increase national awareness of the problem of AMR and the second was to improve knowledge of AMR and related topics.

An investigation into health professionals’ KAP on AMR in Sri Lanka showed that approximately 40% of trainee nurses thought that taking antibiotics would help to prevent colds. Sakeena et al. [9,10] investigated Sri Lankan pharmacy students’ knowledge of antibiotics and AMR and conducted a comparative study with the knowledge of Australian pharmacy students, demonstrating that AMR knowledge of Sri Lankan pharmacy students was less than their Australian counterparts. Tillekaratne et al. [11] reported a qualitative study that was conducted to investigate the attitude of Sri Lankan physicians’ towards acute respiratory tract infection diagnosis and treatment. The authors found that more than 70% of patients received prescriptions for antibiotics, and the key reasons for over-prescription were high patient volume, diagnostic uncertainty, concern for bacterial superinfections and antibiotic-demanding behavior of patients. During the behavior, patients requested the antibiotic by name or by using the term “capsule”.

According to the WHO; “the use of drugs to treat self-diagnosed disorders or symptoms or the intermittent or continued use of a prescribed drug for chronic or recurrent disease or symptoms” is called self-medication [12]. A study in 2017 identified that in the Colombo district of Sri Lanka, the prevalence of self-medication with antibiotics was 6.8%. The practice of self-medication was affected by demographics such as age, but this level was considered low, due potentially to easy access to hospitals [13]. More recently, Zawahir et al. [14] have explored how the general public obtain antibiotics over the counter in Sri Lanka, noting that roughly 30% of pharmacists and assistants supplied antibiotics without a prescription for common infections. While these articles indicate that the general public in Sri Lanka wishes to, and can, obtain antibiotics quite easily, in order to develop appropriate methods to reduce misuse, more research is required to understand what they understand about antibiotics 

Although Sri Lanka is a small island, culturally different communities exist within it. Further, the beliefs, levels of education, income, and the healthcare that the communities receive differ to a great extent [15]. For example, rural communities mostly use traditional herbal medicine as a primary approach, whereas urban communities are prone to using Western medicine. However, due to globalization, there is a high possibility for all communities, including the indigenous community, to receive Western medicine [16,17]. 

It could be hypothesized that the socioeconomic, cultural, and educational background of a community could affect the KAP on antibiotics and AMR in community members, just as demographics such as age had significant effects on the practice of self-medication in the study by Senadheera et al. [13]. Therefore, the current study aimed to explore the KAP of antibiotics and AMR, alongside perceptions of personal antibiotic usage, of the general public from two distinctly different communities in Sri Lanka; an urban community and a socio-culturally distinct rural community. This study was conducted as part of a larger project investigating the potential role of wildlife in AMR and ecosystem contamination in Sri Lanka.

## 2. Results

A total of 210 self-administrated questionnaires were returned from the urban community, giving a response rate of 71.2%. However, 28 questionnaires were excluded because answers to questions 1 and/or 2 were missing, resulting in 182 questionnaires included within the analysis, with a 7.0% margin of error. In the rural community, the survey was not self-administered and was instead conducted by a researcher interviewing subjects. The response rate in the rural community was 100% because all households that were visited agreed to take part in the survey. However, three participants were excluded since they were reluctant to give answers to many questions including question 2, and therefore 147 questionnaires were taken into the final analysis.

### 2.1. Participant Demographics

The demographic characteristics of the respondents are shown in Table 1. Percentages of age groups were similar in both communities with approximately 50% of respondents within the age group 18 to 40. The percentages of participants’ sex differed in the two areas, with more females in the urban community and more males in the rural community. Education also differed, whereby 31 (21.1%) respondents in the rural community had not attended any school and 41 (27.9%) had only primary education, whereas in the urban community, these values were 0 (0.0%) and 8 (4.5%), respectively. Further, 22 (12.4%) respondents in the urban community had higher education, while 0 (0.0%) rural participants had higher education.

### 2.2. Respondents’ Knowledge Compared to Demographics

#### 2.2.1. Participants’ Thoughts about Their Own Knowledge of Antibiotics

The responses regarding perceptions of knowledge about antibiotics are shown in Figure 1. The majority of respondents in the urban community, 126 (69.2%), believed that they had very good or good knowledge about antibiotics, whereas the majority of respondents in the rural community, 88 (59.8%), believed that they had poor or very poor knowledge.

#### 2.2.2. Identification of Antibiotics by the Participant

Table 2 shows the results of correct and incorrect answers provided by participants when asked to identify antibiotics from a list of 10 medicines. More respondents from the urban community correctly identified each of the antibiotics than from the rural community. However, 60 (32.9%) urban respondents could not identify any antibiotics, and only 13 (7.1%) identified all five. In comparison, 127 (86.3%) respondents in the rural community could not identify at least one antibiotic and no participant was able to identify all five. Many respondents thought that non-antibiotic medicines were antibiotics, for example, Panadol was selected by 39.0% of urban and 100% of rural respondents and Paracetamol by 53.3% of urban and 52.3% of rural respondents (Table 2).

#### 2.2.3. Relationship between Respondents’ Perceptions of Their Own Knowledge and Their Identification of Commonly Used Medicines by Name

When considering the urban community, there was no significant correlation between respondents’ perceptions of their own knowledge of antibiotics and the total number of drugs correctly identified as an antibiotic (*p*-value = 0.057) or the total number of drugs incorrectly identified as antibiotics (*p*-value = 0.840). Respondents in the rural community had a statistically significant moderate-to-strong correlation between respondents’ perceptions of their own knowledge of antibiotics and the total number of drugs correctly identified as an antibiotic (r = 0.49 and *p*-value = 0.001) and the total number of drugs incorrectly identified as an antibiotic (r = 0.78 and *p*-value = 0.001).

#### 2.2.4. Association between Respondents’ Perception of Their Own Knowledge and Demographic Characteristics

Respondents who rated their knowledge of antibiotics as very good or good were grouped (Good), and poor or very poor were grouped (Poor). There was no significant association between respondents’ perceptions of their own knowledge of antibiotics and their socio-demographic characteristics in terms of gender (*p*-value; urban: 0.584 and rural: 0.99) or age (*p*-value urban: 0.91 and rural: 0.41) in both areas. However, respondents in the rural area had a statistically significant association between respondents’ perception of their own knowledge of antibiotics and their highest education level (*p* = 0.001) (Table 3). Moreover, if the respondents had above school-level education, the odds of being able to select the correct antibiotic from the given list increased by a factor of 3.108 (95% confidence interval from 1.200 to 8.049) over the respondents who had school-level education only. 

#### 2.2.5. Association between Respondents’ Demographic and Ability to Select Antibiotics 

There were no respondents from the rural community that had received higher education (post-school-level, i.e., university or HE college), so results for this community were analyzed for primary and higher school (up to O/L General Certificate of Education (Ordinary Level) (average age 15–16 years) and up to A/L General Certificate of Education (Ordinary Level) (average age 17–19 years)).

Respondents who selected less than three drugs as antibiotics were grouped as having poor ability to select antibiotics and respondents who selected three or more than three out of five were grouped as having good ability to select antibiotics. A total of 48 (26.4%) respondents in the urban community had good ability to select antibiotics from the given list, whereas no one had good ability to select antibiotics in respondents in the rural community. Hence, only respondents in the urban community were taken into the further analysis. Level of education showed a significant association with respondents’ ability to select antibiotics; respondents who had higher education showed good ability to select antibiotics from the given list of different medicines (*p* = 0.008). However, there was no significant association between respondents’ demographics in terms of gender or age and the ability to select antibiotics (Table 4).

The following results in Section 2.3, Section 2.4 and Section 2.5 and Section 2.6 relate to the participants’ own understanding of antibiotics. As the above results have demonstrated, this is often inaccurate compared to the true definition, and should be interpreted as such.

### 2.3. Respondents’ Knowledge of Antibiotic Effectiveness and Action 

Only 32 (18.3%) urban participants and 19 (12.9%) rural participants correctly identified that antibiotics are only effective against bacteria (Figure 2). In addition, many respondents in both areas mistakenly thought that a variety of ailments can be treated with antibiotics (Figure 3), which aligns with their misunderstanding of what is classified as an antibiotic.

### 2.4. Respondents’ Perception of Antibiotics 

As shown in Table 5, respondents’ perception regarding appropriate antibiotic usage were evaluated using six statements. The majority of rural respondents, 123 (83.7%), thought that a pharmacist is capable of prescribing antibiotics, and 58 (39.5%) thought that it is acceptable to buy antibiotics for animals without advice from a veterinary doctor. These values were 32 (18.5%) and 26 (15.2%) in the urban community, respectively.

### 2.5. Perceptions Regarding Personal Usage of Antibiotics

Many respondents (urban—33.6% and rural—85.1%) thought that they had taken antibiotics within the last month or within the past six months, according to their own understanding of the term. This shows that a very high proportion of people in rural areas take “antibiotics” or medicines for aliments. 

### 2.6. Knowledge and Perception of Antimicrobial Resistance (AMR)

In this section, respondents were asked to rate their knowledge of AMR. In the urban area, 89 (51.4%) though that they had good knowledge regarding AMR while 84 (48.6%) respondents thought their knowledge was poor. However, all participants in the rural area reported that they did not know about AMR. 

The majority of the respondents in the urban area, 132 (75.4%), reported that they acquired their knowledge of AMR from doctors, and 50 (28.7%) reported that they used mass media. 

Full details of participants’ knowledge and perception regarding AMR can be seen in Figure 4 and Figure 5. Half of the respondents thought that AMR is only a problem for people who take antibiotics regularly, and a similar percentage of respondents thought that bacteria that resist an antibiotic cannot transmit from person-to-person.

## 3. Discussion

Misuse of antibiotics by members of the public is likely to be one cause of the development of AMR. To understand this potential problem, population-based studies regarding antibiotic usage by members of the public and AMR have been conducted in several countries, but little is known about this issue in Sri Lanka [7,18,19]. We think that the current study is the first survey to explore public perception and knowledge about antibiotics and AMR as well as perceived antibiotic usage in Sri Lanka. 

The majority of the urban respondents in our study believed that they had “very good” or “good” knowledge regarding antibiotics, but in the rural community, over half of the respondents self-reported “poor” or “very poor” knowledge. Although many urban respondents thought their knowledge to be at a high level, this did not significantly correlate with their actual ability to identify antibiotics. A similar finding has been reported from a survey in Lithuania, where members of the public overestimated their knowledge about antibiotics and their correct usage [20]. The mismatch between self-reported knowledge and actual knowledge about antibiotics is likely to be an important factor in the rise of AMR and the difficulty in combatting this global problem. 

The perception of one’s own knowledge may play a decisive role in shaping cognition and influencing one’s decisions and behavior and individuals may be less inclined to systematically seek information on a topic they believe they are well-informed about [21]. 

A recent study was conducted to see how perceived threat to an individual relates to their media use, and how media use relates to perceived and actual knowledge of COVID-19 [22]. The results depicted that people who felt the most threatened by the COVID-19 pandemic used media platforms more often to obtain information. In our study, 52.5% of urban respondents were not worried about the impact of AMR on their lives, and as such, may not seek information on this topic, despite a potential lack of knowledge. However, it should be noted that within the COVID-19 study, those respondents who were threatened and used social media focused on fewer media channels, and while frequent use of the media was associated with greater perceived awareness, it was not associated with greater actual awareness of COVID-19 (i.e., illusion of knowledge), while the use of fewer media channels was associated with greater perceived and actual knowledge [22]. It is therefore important to not only consider how to encourage the public to seek knowledge when required, but to support and facilitate their search for the correct information. 

In the current study, the rural community showed poorer ability to select antibiotics by their name than the urban community. For example, in this study, 57.1% of urban respondents, and virtually all rural participants (98.6%) were unable to identify amoxicillin correctly, which is a frequently used and popular antibiotic. Despite this evident lack of knowledge, amoxicillin was the most correctly identified antibiotic by urban respondents, while penicillin was the most correctly identified by rural participants. Amoxicillin and penicillin were also the most-identified antibiotics in a study of another South Asian country, Bhutan. However, in that study, more participants successfully identified amoxicillin (70%) and penicillin (43%) than in the current study [23]. In both Bhutan and in our results, 30–40% of respondents incorrectly identified paracetamol as an antibiotic [23]. Within our rural community, non-antibiotics were actually more often identified as antibiotics than the antibiotics. For example, only 10.8% identified penicillin as an antibiotic, while 100% identified Panadol as an antibiotic. In Sri Lanka, painkillers such as Panadol are sold widely, including in small grocery shops in villages, whereas antibiotics are only sold in a pharmacy. Television commercials, newspaper advertisements and posters also display the value of Panadol for symptomatic treatment. Therefore, respondents might be more familiar with the term Panadol than the name of any other medicines and they may therefore mistakenly choose painkillers such as Panadol as an antibiotic. These findings are especially relevant for policymakers at national and international levels. While the public may be aware of the word “antibiotics”, given that most cannot identify them correctly, awareness campaigns should be careful in the way that they present advice, and should not assume knowledge. 

The general public in areas such as Sri Lanka therefore may not possess sufficient knowledge to categorize medicines as “antibiotics” by their name. However, names are not the only way to identify antibiotics, and a study in Mozambique reported that even though the participants did not have an accurate understanding or knowledge of antibiotics, they were able to identify amoxicillin capsules by their distinct red and yellow colors [24]. Future studies and policymakers should consider the possible importance of non-verbal knowledge regarding the identification and use of antibiotics. 

In Sri Lanka, only 10–12% of the population have post-secondary school/university level or higher education, and 63% have high school level education [25]. The level of education of questionnaire respondents showed a significant association with the ability to identify antibiotics by name in the urban community, while gender and age did not. There are only a limited number of studies that explicitly compare demographics and antibiotic knowledge, for an example, Waaseth et al. [26] reported that younger individuals had less knowledge of antibiotics than older individuals. Recently, a study by Effah et al. [27] reported that working in the healthcare sector is a major contributor to the level of knowledge of antibiotic resistance. Other medical-related topics show similar results, for example, knowledge regarding breast cancer can depend on the level of education of an individual [28]. This suggests that public awareness programs to improve the use of antibiotics should be tailored to different educational levels, with a particular focus on those who only possess school-level education. 

The majority of respondents in our study incorrectly replied that antibiotics are effective against viruses, in parallel with previous findings from Sri Lanka and other countries such as Japan [18,29,30,31]. It may be that there is confusion over not just “antibiotics”, as indicated in this study, but over words such as “viruses” too. In the Sri Lankan context, the authors anecdotally note that, instead of using specific words such as “virus” and “bacteria”, many health education programs use the word “microbes” or “germ” to introduce pathogens to the public. Therefore, the public may lack the ability to differentiate them by their exact names. When compared with the overall results in a WHO multicounty survey, Sri Lankan people are also less likely to know that antibiotics are not effective against cold and flu [6]. In addition, the results in this study regarding diseases and symptoms treatable by antibiotics were lower compared to a study in an urban area in Senegal [32]. Some of our respondents mistakenly thought that painkillers were antibiotics, which may explain why respondents identified that headaches and body aches could be treated with antibiotics. This unpacking of the participants’ misunderstanding of antibiotics, and which ailments they can treat, could be used to develop bespoke training for different communities within Sri Lanka.

Most respondents thought that they had taken antibiotics within the past 6 months. However, this may not be the actual usage, as many respondents in our study thought that non-antibiotics such as paracetamol were antibiotics. In the urban community of our study, the reported proportion was lower compared to reports in the WHO multi-country survey in 2015, whereas it was higher in the rural community [6].

Almost all rural community members in our study stated that they were not aware of AMR at all. The majority of the respondents in the urban community in our study (68%), compared to 75% of respondents in a Nigerian study and 85% in a Swedish study, assumed that antibiotic resistance occurs when their body becomes resistant to antibiotics [19]. A similar proportion of respondents in the urban community in our study and respondents in the Nigerian study reported that bacteria which are resistant to antibiotics can spread from person-to-person. Most respondents in the urban community in our study believed that they are not at risk of developing antibacterial resistance to infection if they take antibiotics correctly. Based on literature from different parts the world, on average 56% of respondents in higher-income countries and 71% of respondents in lower-income countries have reported the same misconception [6]. These studies identify useful areas where knowledge needs to be transferred from scientists to the public in a way which is accessible, considerate of the public’s needs, and makes use of trusted sources of knowledge [33].

There were several limitations in our study. First and foremost, due to the COVID-19 outbreak, we were unable to achieve the desired sample size. Therefore, the margin of error of the results was increased and the conclusions should be understood in light of this. In addition, the study was conducted in an area of one square kilometer in each community. Hence, we are unable to generalize and apply the results to the whole of Sri Lanka; however, the similarity of our results to studies in other countries increases the confidence that they are applicable to more than these two isolated locations. Although we used the same questionnaire, we could not follow exactly the same methods to collect data from both communities, and instead used methods which would elicit the greatest responses in each locality. In the urban community, a self-administered questionnaire was used, whereas in the rural community, responses were given in-person to a researcher. This may have led to differences in response such as participants in the presence of a researcher responding in a way which was seen as favorable, or participants completing the self-administered questionnaire looking up answers. While the results of the two communities are presented here for an understanding of different locations within Sri Lanka, results have not been statistically compared in part due to these methodological differences. The second half of the questionnaire required participants to describe their use of antibiotics. Given their lack of understanding of what an antibiotic is, identified in the first part of the questionnaire, these results have to be interpreted appropriately. While they allow us to understand participants’ perceptions of their own antibiotic use, for example, they do not accurately depict antibiotic use in Sri Lanka. Future research may therefore be directed to include only individuals who are known to have been prescribed antibiotics; however, this was outside the possibilities of the current research. Alternatively, methods whereby a definition of antibiotics is provided and understanding is checked could be used before a respondent completes a questionnaire on antibiotic use, in order to ensure participants’ understanding of the questions and researchers’ interpretations of the results. Even though we asked participants to fill out the form independently; there may be a chance that participants completed the form while discussing it with other family members. As such, it is a limitation of the study that participants within a household were treated as independent responses. Due to the anonymity of submitting the questionnaires, we were unable to identify responses from the same household and thus could not use statistics taking this into consideration; however, this approach was used as a means to encourage individual responses.

Despite the limitations, our study is a useful first step to identify the gaps in the knowledge and understanding of antibiotics and AMR in Sri Lanka, which may be used to guide further research to identify how to best educate these diverse populations. 

## 4. Materials and Methods

### 4.1. Study Setting

This study aimed to investigate two locations, chosen due to their contrasting features, especially their urban or rural location. One sq km area from each of the two following community areas was selected as the study sites.

#### 4.1.1. General Urban Community

The urban community was selected from a district in the Western province in Sri Lanka. This district has the highest literacy rate (95.4%) out of 18 districts, and the second highest population density, at 1539/persons per sq km [15]. The study site included approximately 1200 households.

#### 4.1.2. Rural Indigenous Community 

The rural community was selected from a district of Uva province in Sri Lanka. This district has a lower literacy rate (85.2%) the 15th out of 18 districts, and the population density is around 279/persons per sq km [15]. The study site included approximately 125 households. The study site is located in an area where an indigenous tribal community lives. There is an attempt to maintain this indigenous community and their cultural practices for the preservation of ancient traditions including indigenous medicine usage.

### 4.2. Questionnaire Development 

A questionnaire was developed as the data-collection tool (Appendix A). It consisted of 24 closed and one open-ended question. It incorporated four sections: thoughts on antibiotics (3 questions), personal usage (11 questions), thoughts on AMR (6 questions), and demographic characteristics (5 questions). 

The first question asked respondents to rate their knowledge regarding antibiotics on a scale of “very good”, “good”, “poor” or “very poor”. The second question asked participants to identify antibiotics from a list of 10 commonly known medicines in Sri Lanka, used in self-medication (either generic or brand names), which consisted of five correct answers and five incorrect answers. The answer options were created from the pilot survey (information below). The five most commonly mentioned antibiotics and five most commonly mentioned non-antibiotics from an open-ended question were added together to make the list of 10 medicines. This question format was based on other surveys which attempted to identify participants’ awareness of an issue, such as horse owners’ awareness of exotic diseases [34]. The third question in this section allowed participants to name any other antibiotics they knew. Questions 1 and 2 of this section form the basis of this study regarding the general public’s knowledge of antibiotics in Sri Lanka. While these questions only address one aspect of identifying antibiotics, by a specific name, rather than visually or potentially by locally known names, it represents a first step in unpacking the KAP regarding antibiotics of the general public in Sri Lanka. Responses to other sections of the questionnaire are also included where appropriate, as described below in the analysis section. 

The questionnaire was developed in English by subject experts to ensure its content and relevance, and then translated into Sinhalese. It was partly based on pre-existing questionnaires which have been used globally to explore perceptions of AMR including the WHO multi-country survey which was conducted in 2015 [6]. The questionnaire was piloted with 10 participants from each study site mirroring the methods in the study, namely, a self-administrated pilot in the urban setting and an interviewer-administered pilot in the rural setting. Results of the pilot test were analyzed to identify any potential flaws in the questionnaire design. Minor adaptations were made based on these results. Adaptations were made to question 2, which was tested as an open-ended question but finally converted into a closed question. After considering the pilot study of the rural indigenous community, one question (question number 16) was altered by adding “I do not know” to the answer.

In the urban community, the questionnaire was self-administrated on paper. However, a self-administered questionnaire would have been a limiting factor to gaining sufficient responses in the rural community due to low literacy rates and language barriers. Therefore, the survey was conducted face-to-face by two interviewers who were from the rural community. The interviewers were trained by the main author who had previous experience in conducting surveys in this population.

### 4.3. Sample Size

The required sample size was calculated by Cochran’s formula to estimate population proportion under simple random sampling. A confidence level of 95% and desired marginal error of 5% were chosen. The population size within the one sq km was taken as 1250 in the urban area and 245 as in rural area. The calculated sample size was cross-checked with sample size obtained from the Raosoft online sample-size calculator [35]. The total required sample size was obtained as 295 and 150 for urban and rural areas, respectively. 

### 4.4. Data Collection 

To identify the participants required for the study, simple random sampling was conducted. Random numbers were generated to select a household from the electoral register for the year 2019.

Initially, 300 questionnaires were distributed to 95 households in the urban area. Further distribution was intended based on initial response rates, however, due to the COVID-19 pandemic, this was not possible.

The questionnaire was distributed with the help of the government officer (“Samurdhi” development officer) and consent to participate was obtained from the head or the second responsible person from each household. Within a household, only those who were over 16 years old, permanently living in the selected area, and who could listen, speak, read and write were included in the study. Participants were advised to take part in the survey alone without sharing answers with other family members and asked to complete the survey without searching for answers via the internet or other sources. Questionnaires were collected with the help of the government officer, and those who did not respond within two weeks despite two reminders were considered as non-respondents.

In the rural community, two freelance interviewers (1 female and 1 male) with experience regarding the indigenous culture and language were employed to conduct the survey. Upon visiting the randomly selected household, only those who were over 16 years old, permanently living in the selected area, and who could listen and speak were included in the study. Approximately 10 participants were interviewed per day until the required number of participants had taken part.

### 4.5. Analysis 

Incomplete responses, whereby data for the first two questions were missing, were removed from subsequent data analysis. 

All responses were numbered before the analysis and data were entered into an Excel spreadsheet. Statistical analysis was performed using SPSS version 27 software. Data obtained from Section 1 and Section 2 were summarized using descriptive statistics, and the total number and percentage were calculated. 

Spearman’s rank correlation test was used to explore perceived and actual knowledge. The test was conducted to identify the correlation between knowledge rated by respondents (Question 1) and the total number of correct answers given and the total number of incorrect answers given by the respondent (Question 2), at a 0.05 significance level.

The chi-squared test (or Fisher’s exact test, where appropriate) was used to determine the association between respondents’ demographics (age group, gender, education) and their thoughts regarding their knowledge and actual knowledge (ability to select antibiotics), at a 0.05 significance level. To determine the association between respondents’ thoughts on their knowledge or identification of antibiotics with their sociodemographics, the age groups “16 to 18” and “19 to 40” were considered as the young age groups, while “41 to 60” and “more than 60” were considered as the old age group. Regarding education level in the urban community, respondents who identified their highest education level as either: up to primary school, General Certificate of Education (GCE), Ordinary Level (O-level) or GCE Advanced Level (A-level) qualification were grouped as “school-level education” while university and post-secondary school diploma levels were considered as “higher education”. In the rural community, no respondents had received higher education. Hence, respondents who had not attended school and who identified their highest education level as up to primary school were grouped as “lower school level education”, and respondents who had General Certificate of Education (GCE), Ordinary Level (O-level) and GCE Advanced Level (A-level) qualifications were grouped as “upper school- level” education. 

Responses from the two communities are presented in order to understand the KAP in different communities within Sri Lanka. However, the results are not statistically compared between the two communities, as the methods used to collect the data differed, and were chosen to elicit the greatest number of responses in both locations

### 4.6. Ethical Consideration

This study was approved by the ethics review committee, Faculty of Medicine, University of Peradeniya, Sri Lanka. Ethics grant No. 2020/EC/28.

## 5. Conclusions

Based on our findings, it can be tentatively concluded that both urban and rural community members of the public in Sri Lanka have a poor ability to identify antibiotics from a list of commonly used medicines and poor knowledge regarding the appropriate use of antibiotics and understanding of AMR. Therefore, it is important to enhance awareness of what is meant by “antibiotics”, antibiotic effectiveness, correct antibiotic usage, disposal and AMR amongst the general public. This can likely be achieved through locally relevant educational interventions and/or communication initiatives, such as through social media, television, radio, newspapers and posters targeted towards those with different educational backgrounds, within the communities of Sri Lanka.

## Figures and Tables

**Figure 1 antibiotics-11-00454-f001:**
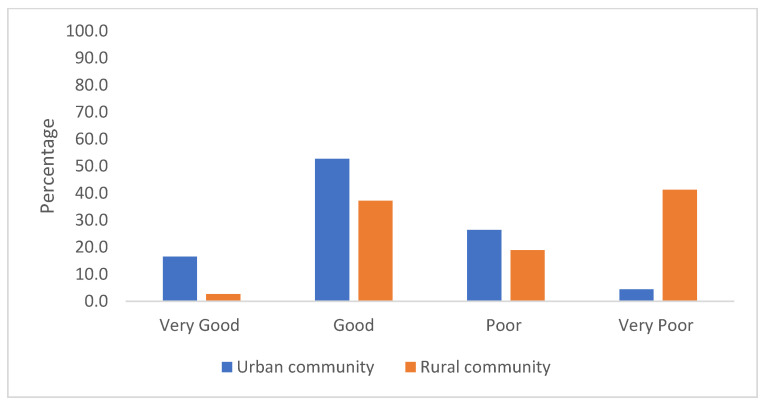
Respondents’ perceptions of their knowledge regarding antibiotics.

**Figure 2 antibiotics-11-00454-f002:**
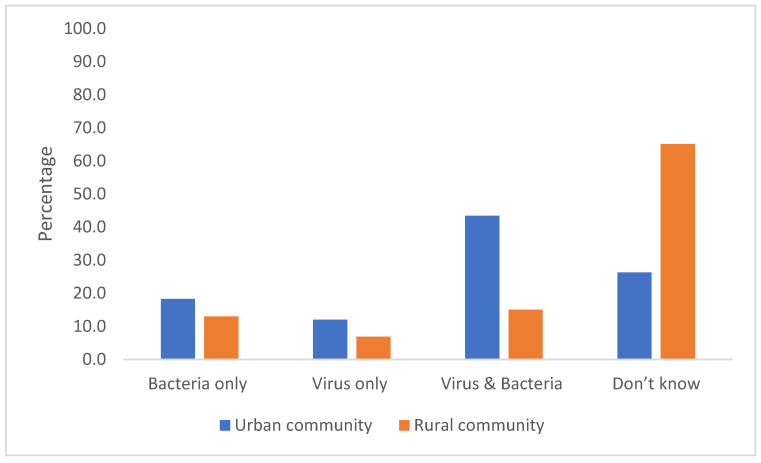
Responses of urban and rural respondents regarding the question “Which of the following do you think antibiotics are effective against?”.

**Figure 3 antibiotics-11-00454-f003:**
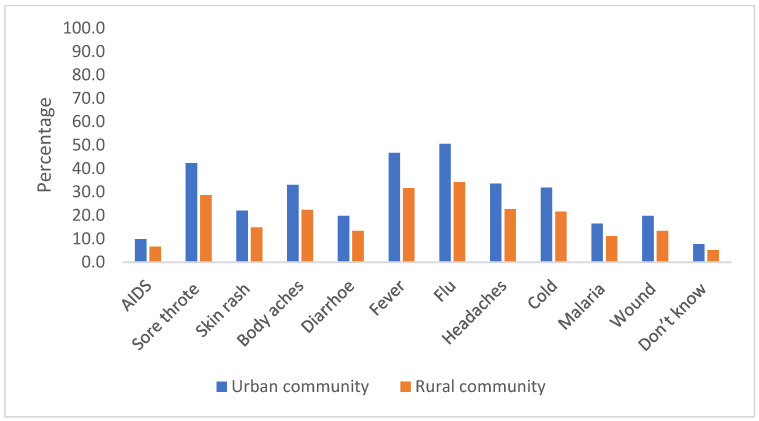
Respondents’ opinions of which diseases or symptoms could be treated with antibiotics.

**Figure 4 antibiotics-11-00454-f004:**
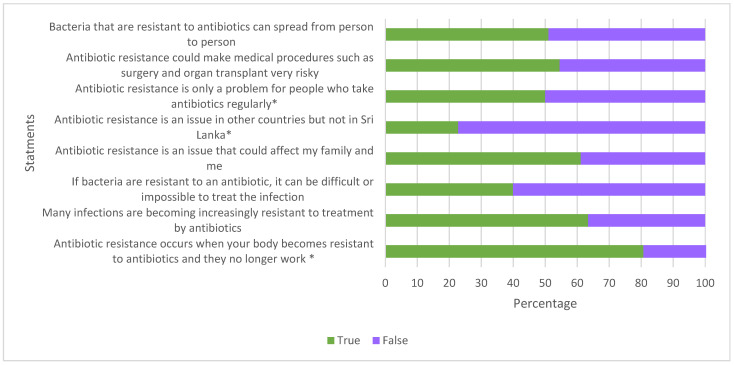
Urban respondents’ knowledge of AMR. Answers of “true” indicate good understanding of AMR, apart from for negatively worded statements (highlighted by *) where an answer of “false” would indicate knowledge of AMR.

**Figure 5 antibiotics-11-00454-f005:**
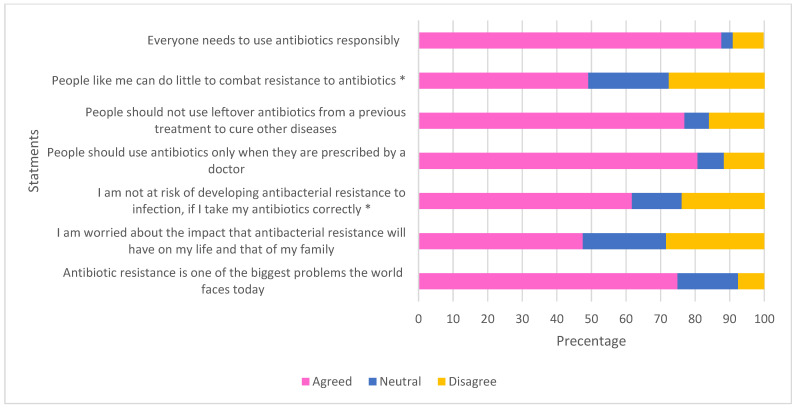
Urban respondents’ perception of AMR. Answers of “agreed” indicate good understanding and attitudes towards AMR, apart from for negatively worded statements (highlighted by *), where “disagree” indicates good understanding and attitudes towards AMR.

**Table 1 antibiotics-11-00454-t001:** Demographics of respondents in relation to the site, gender, age group, and highest education level.

Characteristics	Range/Group	Urban Community *n* (%)	Rural Community *n* (%)
Sex	Female	132 (73.3)	61 (41.2)
	Male	48 (26.7)	83 (56.1)
	No schooling	0 (0.0)	31(21.1)
Education level	Primary Education only	8 (4.5)	41 (27.9)
	Up to O/L General Certificate of Education (Ordinary Level) (Average age 15–16 years)	84 (47.5)	67 (45.6)
	Up to A/L General Certificate of Education (Ordinary Level) (Average age 17–19 years)	63 (35.6)	8 (5.4)
	Higher education post-school level i.e University or HE College)	22 (12.4)	0 (0.0)
Age group	Young (16–40 years)	98 (55.6)	88 (59.8)
	Old (41 to 60 years and above)	78 (44.4)	59 (40.2)

**Table 2 antibiotics-11-00454-t002:** Percentages and number of respondents who correctly identified the five antibiotics, and how they incorrectly identified the other five drugs as antibiotics from the given list of 10 commonly used medicines.

Identification	Name of the Medicine/Drug	Urban Community *n* (%)	Rural Community *n* (%)
Identified antibiotic correctly	Amoxicillin	78 (42.9)	2 (1.4)
	Penicillin	65 (35.8)	16 (10.8)
	Ampicillin	55 (30.2)	5 (3.4)
	Streptomycin	51 (28.0)	0 (0.0)
	Tetracycline	35 (19.2)	0 (0.0)
Identified other drugs/medicines as antibiotics	Paracetamol	97 (53.3)	77 (52.3)
	Piriton	73 (40.1)	48 (32.4)
	Panadol	71 (39.0)	147 (100)
	Aspirin	42 (23.1)	19 (12.8)
	Folic acid	25 (13.7)	3 (2.0)

**Table 3 antibiotics-11-00454-t003:** Respondents’ perceptions of their own knowledge regarding antibiotics, compared between the demographics of gender, age, and level of education of respondents.

Community	Characteristics	Range/Group	Respondents’ Thoughts on Their Own Knowledge		χ2	*p*
			Poor *n* (%)	Good *n* (%)		
Urban Community	Gender	Female	86(67.7)	41(32.3)	0.442	0.584
		Male	35(72.9)	13(27.1)		
	Age group	Young	68(70.1)	29(29.9)	0.012	0.910
		Elder	52(69.3)	23(30.7)		
	Level of Education	Up to School education	105(69.1)	47(30.9)	0.294	0.796
		Higher education (post-school level, i.e., University or HE College)	15(75.0)	5(25.0)		
Rural community	Gender	Female	38(62.3)	23(37.7)	0.000	0.990
		Male	51(62.2)	31(37.8)		
	Age group	Young	50(57.5)	37(42.5)	0.760	0.410
		Old	38(64.4)	21(35.6)		
	Level of Education	Primary school	55(76.4)	17(23.6)	14.830	0.001
		Higher School	34(45.3)	41(54.7)		

**Table 4 antibiotics-11-00454-t004:** Association between respondents’ demographics and ability to correctly select antibiotics from a given list of medicines, by the urban community.

Characteristic	Range/Group	Ability to Select Correct Antibiotics		χ^2^	*p*
		Poor *n* (%)	Good *n* (%)		
Gender	Female	93 (74.0)	34(26.0)	0.347	0.575
	Male	33(70.8)	15(29.0)		
Age group	Young	70(73.2)	27(26.8)	0.047	0.866
	Old	53(72.0)	22(28.0)		
Level of education	School education only	114(75.7)	38(24.3)	7.808	0.008
	Higher education (post-school level, i.e., University or HE College)	9(49.0)	11(51.0)		

**Table 5 antibiotics-11-00454-t005:** Respondents’ perception regarding antibiotic usage.

	Urban Community		Rural Community	
Statements	True *n* (%)	False *n* (%)	True *n* (%)	False *n* (%)
It is advised to use antibiotics that were given to another person if the antibiotics are used to treat the same symptoms or illness	31 (17.8)	143(82.2)	40(27.2)	107(72.8)
It is acceptable to buy the same antibiotics, without consulting a doctor or pharmacist, if you are sick and they helped you to fight the same symptoms in the past	21(12.0)	154(88.0)	27(18.4)	120(81.6)
It is good to keep leftover antibiotics at home in case of the future need	32(18.5)	141(81.5)	52(35.4)	95(65.6)
A prescription from a doctor is needed to purchase antibiotics	152(88.4)	20(11.6)	142(96.6)	5(3.4)
A pharmacist is capable of prescribing antibiotics	32(18.5)	141(81.5)	123(83.7)	24(16.3)
it is acceptable to buy antibiotics for animals without advice from a veterinary doctor	26(15.2)	145(84.8)	58(39.5)	89(60.5)

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
