# Peer review of "Misconceptions of Antibiotics as a Potential Explanation for Their Misuse. A Survey of the General Public in a Rural and Urban Community in Sri Lanka"

_antibiotics, 2022, doi:10.3390/antibiotics11040454_

Round 1

Reviewer 1 Report

This is nicely written and presented manuscript. Thanks for addressing my comments in a previous submission. I have no more comments. 

Author Response

Thank you very much for your valuable comments. Please see the attachment

Reviewer 2 Report

Thank you for giving me the opportunity to review the article. The author conducted a study focusing on the misconceptions of antibiotics as a potential explanation for their misuse. The topic is socially important, but there are crucial problems in the manuscript. Therefore, the reviewer thought that the manuscript cannot be accepted for publication in the journal. I listed major comment below.

Comments:

Title:

  1. The relationship between “misconceptions of antibiotics” and “misuse” is difficult to estimate from this study.

Materials and Methods:

  1. There are several unclear points in the questionnaire development process. The appropriateness of the “a list of 10 commonly known medicines in Sri Lanka” and “the general public’s knowledge of antibiotics in Sri Lanka” are unclear (there are no references).
  2. The process of this study is problematics. The authors conducted different way in urban and rural community. The results cannot compare directly.

Discussion:

  1. As the authors mentioned, the generalizability of this study Is limited.

Author Response

Thank you very much for your valuable comment. Please see the attachment 

Reviewer 3 Report

In my opinion the article has improved, the methods are more transparantly described and the results are put into context now.

I appreciate new title.

Author Response

Thank you very much for your valuable comments 
